# The Role of Conventionally Fractionated Radiotherapy and Stereotactic Radiotherapy in the Treatment of Carcinoid Tumors and Large-Cell Neuroendocrine Cancer of the Lung

**DOI:** 10.3390/cancers14010177

**Published:** 2021-12-30

**Authors:** Mateusz Bilski, Paulina Mertowska, Sebastian Mertowski, Marcin Sawicki, Anna Hymos, Paulina Niedźwiedzka-Rystwej, Ewelina Grywalska

**Affiliations:** 1Department of Radiotherapy, Medical University of Lublin, Chodźki 7 St., 20-093 Lublin, Poland; mateuszbilski@umlub.pl; 2Department of Brachytherapy, St. John’s Cancer Center, Jaczewskiego 7 St., 20-090 Lublin, Poland; 3Department of Radiotherapy, St. John’s Cancer Center, Jaczewskiego 7 St., 20-090 Lublin, Poland; 4Department of Experimental Immunology, Medical University of Lublin, Chodźki 4a St., 20-093 Lublin, Poland; paulinamertowska@umlub.pl (P.M.); sebastianmertowski@umlub.pl (S.M.); annahymos@umlub.pl (A.H.); ewelinagrywalska@umlub.pl (E.G.); 5Institute of Medical Sciences, Medical College of Rzeszow University, mjr. W. Kopisto 2a St., 35-959 Rzeszow, Poland; marcin.sawicki@interia.pl; 6Institute of Biology, University of Szczecin, 71-412 Szczecin, Poland

**Keywords:** typical carcinoid, atypical carcinoid, small cell lung cancer, large cell neuroendocrine cancer, radiotherapy, neuroendocrine tumors, neuroendocrine carcinomas

## Abstract

**Simple Summary:**

Patients with neuroendocrine neoplasms (NETs) are a rare group of patients, 70% of which are diagnosed in the location of tumors in the digestive system, and the remaining 30% in the respiratory system. Building an appropriate therapeutic strategy in a patient with NET requires the involvement of a multidisciplinary team, which should include: oncology surgeon, clinical oncologist and radiation oncologist. One of the commonly used methods of treating lung NETs is the use of radiotherapy. However, the number of available recommendations for treatment of NET radiotherapy is negligible. This poses a significant problem for radiation oncologists when making qualification decisions for treatment with radiant energy. The aim of this article was to present the current knowledge on the use of radiotherapy in the treatment of lung NETs. In addition, we hope that the description of clinical cases in this publication will help radiation oncologists make the best, often personalized qualification decisions.

**Abstract:**

The occurrence of neuroendocrine tumors among the diagnosed neoplasms is extremely rare and is associated with difficulties in undertaking effective therapy due to the histopathological differentiation of individual subtypes and the scarce clinical data and recommendations found in the literature. The choice of treatment largely depends not only on its type, but also on the location and production of excess hormones by the tumor itself. Common therapeutic approaches include surgical removal of the tumor, the use of chemotherapy, targeted drug therapy, peptide receptor radionuclide therapy, and the use of radiation therapy. This article reviews the current knowledge on the classification and application of radiotherapy in the treatment of lung NETs. Case reports were presented in which treatment with conventional radiotherapy, radical and palliative radiochemotherapy, as well as stereotactic fractionated radiotherapy in the treatment of typical (TC) and atypical (AT) lung carcinoids and large cell neuroendocrine carcinoma (LCNC) were used. We hope that the solutions presented in the literature will allow many radiation oncologists to make the best, often personalized decisions about the therapeutic qualifications of patients.

## 1. Introduction

The current classification of pathological neuroendocrine neoplasms (NENs) in various organs uses a number of terminology and site-specific criteria, which creates inaccuracies and confusion not only among patients, but also among pathologists and treating physicians. Therefore, the World Health Organization (WHO) in cooperation with the International Agency for Research on Cancer (IARC) has decided in recent years to systematize the framework for classifying this type of cancer. Two terminologies have been proposed within the NEN classification: the term NEC, which clearly indicates the malignant histology of a high degree and biological behavior of the tumor, and NET (neuroendocrine neoplasm), which refers to a family of well-differentiated neoplasms with the site-dependent potential for metastasis or invasion of adjacent tissues, type and degree of its advancement [1,2,3]. It should be noted that NENs are a group of relatively rare neoplasms, including heterogeneous tumors characterized by the presence of neurosecretory granules with a characteristic histology and immunological profile [1,4]. The statistical data available in the literature show that approximately 70% of the diagnosed cases of NETs concern the location of tumors in the digestive system, and the remaining 30% in the respiratory system [5,6]. The number of diagnosed cases of this type of cancer around the world increases every year. Statistically, in recent years, the incidence of NET has fluctuated within the range of ~2.5–5 cases/100,000 per year, of which the largest proportion are neuroendocrine tumors of the small intestine (carcinoids of the middle intestine) with an incidence of 2.4/100,000 cases per year. However, the autopsy data indicate that the percentage is much higher, and may be as high as 8.4 per 100,000 cases per year [7]. In the United States, there has been a six-fold increase in the incidence of NET over the last 30 years. Data from the SEER database (Surveillance, Epidemiology, and End Results Program) [8] indicate that the incidence in the period 1995–2012 increased from 3.96 to 6.61 per 100,000 cases. These diagnoses were made mainly in Caucasian patients (90.57%), and the dominant gender among whom they were diagnosed was women (54.74%). The disease was dominant in patients aged 50–64 (38% of cases), and in terms of localization, it most often affected the lungs (30.6%), followed by the small intestine (16.82%), rectum and anal canal (11.35%) and colon (9.71%) [9]. On the other hand, studies from Denmark, Sweden, Argentina, France, and Norway indicate the most common location of NETs within the small intestine in 26.9–55.3% of all cases, respectively [10,11,12,13]. The cause of the increase in the incidence of these types of NETs is not fully understood. Scientists suggest that a systematic classification of this type of neoplastic diseases and the development of diagnostic techniques in recent years played an important role in the diagnosis process [14]. The basic tool in the diagnosis of neuroendocrine tumors is the determination of appropriate hormones in the blood. Imaging diagnostics, SRS scintigraphy or Gallium positron emission tomography (PET) are also of great importance (this examination is characterized by a sensitivity of 91% and the highest specificity of 94%). About 80% of neuroendocrine tumors produce a characteristic protein-somatostatin receptor on their surface, which, after administration of gallium-68 (Ga-68) DOTA-peptide, allows us to visualize the tumor and select the appropriate therapeutic strategy [15,16,17,18]. The treatment options for a neuroendocrine tumor largely depend not only on its type, but also on the location and production of excess hormones by the tumor itself. Common therapeutic approaches include surgical removal of the tumor, the use of chemotherapy, targeted drug therapy, peptide receptor radionuclide therapy, or the use of radiation therapy [19,20]. Detailed guidelines for the diagnosis, treatment and management of patients with diagnosed lung neuroendocrine tumors have been included in the recommendations of the European Society for Medical Oncology (ESMO) [21].

Building an appropriate therapeutic strategy in a patient with neuroendocrine neoplasm requires the involvement of a multidisciplinary team, which should include: gastroenterologist, pneumologist, endocrinologist, thoracic surgeons, pathologists, radiologists or nuclear medicine doctors, as well as oncological surgeons, clinical oncologists and radiation oncologists. Only building a comprehensive, multi-level team may allow for the development of the best therapeutic strategy for patients diagnosed with neuroendocrine neoplasms. Relatively few reports concern the use of radiotherapy in this type of neoplasms. The number of available recommendations for the treatment of radiotherapy in NET is negligible. This poses a significant problem for radiation oncologists when making qualification decisions for treatment with radiant energy. The aim of this article was to present the current knowledge on the use of radiotherapy in the treatment of lung NETs. In addition, we hope that this publication will help radiation oncologists make the best, often personalized qualification decisions.

## 2. NET Classification and Characteristics

One of the methods of classifying patients diagnosed with NETs is to determine the secretory functions of the tumor, which may be either present (then these are patients with functional neoplasms that, depending on the type of hormone/hormones secreted, present a wide spectrum of symptoms) or may be absent (then these are patients with non-functional, asymptomatic neoplasms) [22,23]. Currently, the literature distinguishes seven main types of NETs, they are the following subtypes: bronchopulmonary, thymus and those related to the digestive system, jejunum/ileum/colon, duodenum, appendix, stomach, and rectum [24]. Depending on the histology, we can distinguish three NET subtypes according to the WHO classification:G1 neoplasms are well-differentiated, low-grade.G2 neoplasms are well-differentiated and have an intermediate degree of malignancy.G3 are maldifferentiated, high-grade neoplasms.[25] (Table 1).

Within the G1–G3 subtypes, important prognostic factors are the mitotic index, Ki-67 or the presence of necrosis [26,27] (Table 1). The first two factors are related to determining cell proliferation. The mitotic index is determined by counting the mitotic figures in hematoxylin and eosin (H&E) stained samples. Ki-67 is an established marker of cell proliferation that is present at all stages of the cell cycle except the resting phase [28]. The literature data show that the higher the mitotic index and Ki-67 and the more necrotic lesions, the worse the prognosis and the worse clinical course, especially in the context of NETs within the lungs or thymus. There is currently a lively discussion in the literature about updating the cut-off points for both the mitotic index and the Ki-67 index [28].

Based on the results of the NORDIC NEC study involving 252 patients, it is suggested that a higher or equal, i.e., 55%, Ki-67 index should be used to categorize the neoplasm into the high-grade group. Patients with such percentages presented a much better response to platinum-based chemotherapy [31]. The analysis of two other studies on NETs in the gastrointestinal tract indicates that 5% Ki-67 and the mitotic index of 5 mitosis/10 HPF are optimal prognostic factors for the classification of NET as low-grade grade [32,33]. An alternative method of classification is the model based on the TNM scale, i.e., tumor–node–metastasis [34]. Studies have shown that the stage of advancement based on the 8th edition of the TNM classification closely correlates with the 5-year overall survival of patients with NET within the gastrointestinal tract [35]. It applies to 100%, for I-II degree, 91% for III degree, 72% of patients for stage IV with NETs within the small intestine, and 92%, 84%, 81%, and 57% with localization within the pancreas, respectively, for grades I–IV [33,35]; worse prognosis and 5-year survival rate also apply to patients with dissemination to the lymph nodes and distant organs (parameter N and M). The NET classification pathological report should also contain information on the participation of other prognostic factors, e.g., the status of the surgical margin; and vascular or perineural invasion [2,33].

Clinical trials conducted by many research teams have shown that several molecular factors are also involved in disease progression and reduction of therapeutic success, including elevated levels of chromogranin A, overexpression of the mammalian target of rapamycin (mTOR), loss of expression of a cyclin-dependent inhibitor of CDKN1B kinases (p27) or the presence of 1 circulating tumor cell in 7.5 mL of blood (CTC-circulating tumor cells) [2,36,37,38]. The development of molecular techniques also allowed them to support the diagnosis of NETs with a molecular approach and the selection of specialized biomarker molecules with high specificity and sensitivity in the detection of the disease, and involved in the assessment of potential response to treatment. Biomarkers with potential prognostic significance include epidermal growth factor receptor (EGFR), somatostatin receptor (SSR) or programmed death ligand-1 (PD-L1) in pulmonary subtype and glucose transporters type 1 (GLUT-1), O-6 -methylguanine-DNA methyltransferase (MGMT) for pancreatic NETs, as well as connective tissue growth factor for carcinoid heart disease (CCN2) or vascular endothelial growth factor receptor (VEGFR) for gastrointestinal NET [39] (Figure 1).

The use of molecular information in the form of biomarkers can be used not only as an aid in the process of tumor imaging, but also provide valuable information on the pathobiology of the tumor. Additionally, as some studies indicate, this approach can be used by doctors to predict therapeutic efficacy (positive predictive coefficient [PPQ]), which was confirmed in the case of peptide receptor (PRRT) radionuclide therapy [40]. Current NET biomarkers come down to the evaluation of monoanalytes (secretory proteins or amines). Although clinical data show, several of them are effective in diagnosis (e.g., insulin/insulinoma, glucagon/glucagonoma and gastrin/gastrinoma), but alone they account for <2% of all cancers. In addition, the use of biomarkers in diagnostics also has its limitations. At the moment, no single factor has shown sufficient diagnostic or therapeutic utility in estimating the prognosis and predicting response to treatment [41]. That is why research allowing us to evaluate combinations of biomarkers, which will allow us to achieve satisfactory prognostic effects in the future, is so important.

## 3. Classification and Characteristics of Lung Neuroendocrine Neoplasms

There are four basic categories of lung neuroendocrine neoplasms. These are, respectively: typical (TC) and atypical (AT) carcinoids belonging to well-differentiated neuroendocrine tumors (NETs) and small cell lung cancer (SCLC) and large cell neuroendocrine carcinoma (LCNEC) belonging to the poorly differentiated neuroendocrine carcinomas (NECs) (Figure 2).

The most important classification criterion is the assessment of the number of mitoses in the regions with the highest activity in the area of 2 mm². According to the WHO recommendations of 2015, the evaluation in the 2 mm² area is more precise than in the 10HPF [43,44].

The characteristics of these neoplasms are presented in Table 2. Literature data show that each year in the United States, from 2000 to even 4500 cases of lung NETs are diagnosed in adults, which is approximately 1–2% of all lung cancers [45]. TC and AT carcinoids account for approximately 1–2% of all lung tumors [46,47]. AT is the rarest, accounting for 0.1–0.2% of all lung cancers. It is worth noting, however, that they are the most common type of lung cancer occurring up to the age of 20. The vast majority of lung NETs (about 80%) are centrally located [48]. In the previous WHO classification, LCNEC was classified as one of the sub-types of non-small cell lung cancer (NSCLC) and is treated according to the general rules prevailing in this group of cancers. Small cell lung cancer is the most common neuroendocrine neoplasm at this location [49].

The study also shows that the average 5-year survival rate in patients with lung NETs is 89%. However, it depends largely on many factors, including the type of cancer, its location, the occurrence of metastases or the speed and accuracy of the diagnosis. It is this last factor that determines the greatest success, because, as scientific research shows, early diagnosis of this type of cancer allows for a 5-year survival rate of 98%. With the appearance of metastasis, this indicator changes. When local metastases develop, the 5-year survival rate is 87%, and when spread to other parts of the body, the 5-year survival rate is only 58%. The key aspects of an accurate diagnosis include taking into account other factors predisposing the patient to the incidence of this type of cancer, including an accurate family history (a family history of multiple endocrine neoplasms of type 1, which is hereditary, increases the risk of developing lung NETs), race and gender (more common in people with white race, especially women), age (45 years for a typical carcinoid tumor and 55 years for atypical carcinoid tumors) or environmental factors and diet [52] (Figure 3).

The development of neuroendocrine tumors within the lungs can cause a wide variety of symptoms. In the case of the tumor itself, it can block the airways, resulting in coughing or shortness of breath (Figure 3). In functional tumors, hormones secreted by the tumor can cause carcinoid syndrome with a wide range of symptoms and signs not only from the respiratory system, but also from other organs (Figure 3). Although literature data suggest that carcinoid syndrome is more often caused by GI NETs, it should not be disregarded for lung NETs [53]. Currently, most lung NETs are detected unexpectedly during patients’ visits to a physician for reasons not usually related to cancer. When there is a suspicion of this type of disease, doctors recommend a number of tests to confirm and assess the stage of the disease, including: biopsy, bronchoscopy, endobronchial ultrasound X-ray, computed tomography, magnetic resonance imaging, nuclear medicine imaging or blood/urine tests [54]. Due to the fundamental clinical, epidemiological, histological and genetic differences between low and intermediate and high grading neoplasms, there are significant differences in the therapeutic approach to individual types of lung neuroendocrine neoplasms.

## 4. The Role of Radiotherapy in the Treatment of TC and AT Carcinoids

One of the methods used to treat lung NETs is the use of radiotherapy. The process of radiation therapy uses high-energy x-rays or other molecules to kill cancer cells. A radiation therapy regimen or schedule usually consists of a number of treatments given over a fixed period of time at the discretion of the radiation oncologist. Depending on the selected criterion, the literature data classifies radiotherapy in various ways. Taking into account the patient’s clinical condition, we have reactions to three types of radiotherapy:Radical—the highest effective doses of ionizing radiation are used to destroy tumor cells as much as possible [55].Palliative—radiation doses are used in order to effectively relieve symptoms, such as cancer pain, during or after anticancer treatment. It is usually given on an outpatient basis in a clinic or hospital during usually one or 5–10 fractions. Patients treated with this method do not pose a threat to other people because they do not emit radiation [56,57].

Another division of radiation therapy may be classification based on the type of energy used, then we distinguish two types of radiation therapy. The first is orthovoltage radiotherapy using X-rays, which is used to treat, among others, skin cancer [58]. This type of energy is used rather historically. The second and most commonly used is megavolt radiotherapy, which uses gamma or X-rays [58,59]. In order to increase the effectiveness of treatment of cancer patients, very often combination therapy is used. This means strategy involving three basic treatments available, which include surgery, radiotherapy and systemic therapy. The implementation of combination therapy is possible only in a multi-profile and multi-specialist oncology centers [60]. The most common type of radiotherapy for lung NETs is external beam radiation therapy, in which radiation is delivered from an accelerator with a radiation source localized outside of the patient body. It is used as part of a radical or palliative strategy. A helpful tool in making a therapeutic decision based on radiotherapy is the TNM classification, which includes bronchopulmonary carcinoids [61] (Table 3).

### 4.1. Conventional Radical and Palliative Radiotherapy and Radiochemotherapy

In the case of non-surgical treatment of lung carcinoids, we can use radiotherapy, and radiotherapy combined with chemotherapy, as a result of which the patient’s life may be prolonged, as evidenced by tests performed on groups of patients at different stages of advancement. In studies by Wirth et al., four patients in grades IB-IIIB received simultaneous radiochemotherapy. Three patients received chemotherapy based on cisplatin and etoposide and one based on paclitaxel. Atypical carcinoid was diagnosed in three out of four patients. One of the patients with a typical carcinoid tumor, stage IB, after radiochemotherapy, due to disease stabilization, received additional chemotherapy according to the EP regimen (cisplatin with etoposide), after which the disease was completely remitted. Two patients with atypical cancer presented IIIA stage. One of these patients was enrolled in second-line chemotherapy after relapse after previous chemotherapy based on EP chemotherapy with paclitaxel. In both patients, the disease stabilized after radiochemotherapy. In a patient with stage IIIB and atypical carcinoid tumor, a partial response to treatment was observed after radiochemotherapy with paclitaxel. It should be noted that the doses of radiotherapy used in the analysis were relatively low, i.e., 46–54 Gy, which, with potentially lower sensitivity to ionizing radiation in this type of neoplasms, may explain the obtained results—disease stabilization in three patients and partial response in one of them [63]. Another study on a group of 7 patients was presented by Chong et al. among the treated patients, six of them were diagnosed with atypical carcinoid tumor stage IIIA, while the seventh patient was diagnosed with typical stage IIB carcinoid disease. The median follow-up was two years. During this time, two out of seven patients receiving concurrent cisplatin-based radiochemotherapy and etoposide developed [64]. For studies with a larger number of patients, the analyses are based on retrospective studies. One such analysis was carried out by the team of Dasari et al. Researchers analyzed the outcomes of 83 patients diagnosed with typical and atypical lung carcinoids. The results of various treatment regimens in the first and second therapeutic lines were analyzed. The use of EBRT in the first line of treatment concerned 9% of patients, while in 3% of patients EBRT was used as the second line of therapy. In the first line of treatment, one patient received EBRT in combination with octeotride, one with cytotoxic chemotherapy, one with everolimus, in seven EBRT was the only treatment. In the second line of treatment, two patients received EBRT along with octotride and two as sole therapy. This study also addressed the issue of second progression, i.e., the use of a third line of treatment. In a patient who received octotride as a second line of treatment, the time between the first and second progression was 26.7 months. In the third line of treatment, EBRT was used together with octreotide. Another patient, after cytotoxic chemotherapy, experienced a second progression after a month. In the third line of treatment, EBRT was used. Another patient diagnosed with a second progression after EBRT received octreotide therapy on the third line after 5.1 months. A patient who did not receive treatment after the first progression was also described. The time to the second progression was 8.5 months, and EBRT was used in the third line [65]. Unfortunately, the authors of the article did not provide more information on radiotherapy, including the fractionation scheme. Another retrospective analysis is the analysis of Okoye et al. This study retrospectively analyzed the outcomes of different treatment strategies for patients with typical and atypical carcinoids. Of the 63 patients, only three received radical radiochemotherapy as primary treatment and one received SBRT radiotherapy (the results of this patient will be discussed below). After a few months, both patients were diagnosed with progression and underwent emergency surgery. Unfortunately, apart from the information on atypical histology, the authors do not provide the initial stage of the disease or the radiotherapy regimen used in them. The third patient presented atypical histology and initially had three brain metastases. Simultaneous radiochemotherapy according to the 54 Gy scheme in 30 fractions and etoposide with cisplatin were used. Brain metastases were treated by radiosurgery using a gamma knife. One of the lesions was given 20 Gy and the remaining 16 Gy in single fractions. Radiological evaluation using the RECIST criteria, performed at least 3 months after the end of treatment, showed complete remission of the disease in the thorax and stabilization of meta changes in the brain. Unfortunately, the further fate of the patient, after a longer follow-up period, is unknown. A patient was also described, with a grade IIIb and atypical histology, who, after pneumonectomy, received adjuvant radiochemotherapy up to a dose of 59.4 Gy (33 fractions) in combination with temozolamide and capecitabine. After the follow-up period, 26.2 months after the surgery, the disease spread. In patients treated with a palliative assumption, e.g., 35 Gy in 14 fractions, 40 Gy in 20 fractions for thoracic lesions or 24 Gy in three fractions, the disease stabilized after at least three months in the imaging assessment [66]. As indicated by the above data, an effective therapeutic route for this type of disease has not yet been developed, yet some researchers suggest a certain course of action. Researchers from the Center of the Medical University of Freiburg (Kaifi et al.) presented an algorithm of therapeutic management in which external beam radiation therapy (EBRT) is used in the presence of contraindications to surgery, diagnosis of unresectable cancer or recurrence, also as a palliative therapy in the presence of symptoms resulting from advanced disease and the lack of possibility of radical treatment, e.g., due to massive dissemination [67]. In turn, Mackley et al. in their analysis of the use of radiotherapy in patients with lung carcinoids, they recommended its use in all patients with positive surgical margins or after R2 procedures, regardless of the type of carcinoid tumor. In typical carcinoids, after complete resection, regardless of the N feature, they do not recommend adjuvant radiotherapy. In the case of the diagnosis of the N2 feature in atypical carcinoids, according to the authors, adjuvant radiotherapy is indicated, and its role is not clear in patients with the N0/N1 feature. In patients who cannot undergo surgery, radiotherapy is usually recommended. In adjuvant therapy, the median dose was 56 Gy (45–58 Gy) [68]. On the basis of the presented results, although limited, it can be concluded that carcinoid tumors require a higher biologically effective dose to obtain longer, optimally lasting local control of the disease.

### 4.2. The Abscopal Effect in the Treatment of Carcinoids

An interesting issue that was described by Kareff et al. is to obtain an abscopal effect in a patient with typical lung carcinoid [69]. The abscopal effect is the regression of the disease beyond the irradiation area. He is related, inter alia, with re-stimulation, inhibited by tumor-secreted cytokines, of the host immune system. Probably the main cause is activation of CD8^+^ cells (cluster of differentiation 8) [70,71,72]. Previous studies have allowed to record the abscopal effect in several types of malignant neoplasms, incl. gastric cancer, melanoma, lymphoma, kidney cancer [73,74,75,76]. In the aforementioned patient from the work of Kareff et al., the disease was spread to both lungs, three lesions were described in total. After 13 years of use of octeotride, her disease progressed in the form of an increase in the size of the three nodules described in computer tomography (CT). Due to comorbidities, lanreotide was introduced in the next treatment line. During the therapy, there was a further increase in the size of one of the described changes, the so-called oligoprogression. Therefore, the patient was qualified for stereotactic fractionated radiation therapy (SFRT) for a lesion showing progression in the left lower lobe with continuation of lanreotide therapy. A dose of 40 Gy was prescribed in five fractions, administered every other day. From the tomographic images presented in the work, it can be concluded that the tumor had a rather central location defined as a distance of up to 2 cm from the spine of the trachea. Three months after SBRT, a CT scan showed complete regression of the irradiated lesion and of a second non-irradiated nodule within the same lung, the upper lobe. The lesion not irradiated with the same lung decreased from 1.5 cm to 0.5 cm in the greatest dimension. The third nodule in the opposite lung showed dimensional stabilization [69]. Searching the pub-med database, a second case was found that described the achievement of the abscopal effect in a patient with atypical carcinoid tumor. The patient was diagnosed with to the cervical lymph nodes, lungs and subcutaneous tissue. About one year after diagnosis, the patient was enrolled in a clinical trial that included the use of temozolamide and an orally administered PARP (poly ADP ribose polymerase inhibitor) inhibitor. After 4 months of therapy, disease progression was described and systemic therapy was terminated. Due to the presence of symptoms, the patient was qualified for palliative radiotherapy of changes in the cranial vault. A dose of 30 Gy in 10 fractions was used. Regression in the area of metastases to the subcutaneous tissue was observed one month after the end of radiotherapy. Seven months after its completion, imaging tests revealed virtually complete regression of the disease. An assessment after 18 months confirmed that a durable response was obtained [77]. Both of the presented cases relate to the abscopal effect and indicate the systemic action of EBRT. Obtaining this effect, especially after administering the commonly used regimen in palliative treatment, i.e., 10 × 3 Gy, is particularly puzzling. While the use of high fractional doses and radiosurgery accompany most abscopal responses, it is especially in the diagnosis of a carcinoid tumor, a cancer that is believed to have low radiosensitivity, a palliative dose causing a storm effect. This undoubtedly requires further research, the results of which may, in the future, give a chance to cure patients with massive disease spread. Although this phenomenon is extremely interesting from the point of view of the application of therapeutic therapies, it should be mentioned that its incidence is very rare among patients with neuroendocrine tumors of the lungs.

### 4.3. Stereotactic Fractionated Radiation Therapy

The development of radiotherapy technology, including imaging guidance and tumor movement management, has led to more precise delivery of radiation doses to pathological tissue while reducing the amount of radiation affecting healthy cells. Thanks to the above actions, it is possible to safely deliver very high (ablative) doses to smaller tumors without the need for long-term fractionation. This technique, typically used as a cycle of one to five treatments over 1–2 weeks, is referred to as stereotactic body ablative radiotherapy (SABR), and its use in treating medically inoperable stage 1 NSCLC has increased dramatically since 2001r. [78,79,80]. It is commonly believed that low-stage carcinoid tumors require a high biologically effective dose (BED) to optimize outcomes with respect to local disease control. This is due to their low radiosensitivity. Regardless of the type of cancer, low radiosensitivity is currently not a major problem for radiotherapy. It can be overcome by applying stereotactic radiotherapy in a single fraction, the so-called radiosurgery or in several smaller fractions, the so-called stereotactic fractional radiation therapy (SFRT). An excellent example of this is the results of SBRT in patients with low-advanced renal tumors. Research results, including Phase II, in this group of patients with kidney neoplasms commonly considered to be low-radiation sensitive, suggest that this method will soon appear as a therapeutic option in the NCCN recommendations [81,82,83]. The same applies to the use of SBRT in the treatment of carcinoids. The first published study on this issue is the work of Calaco et al. The treatment outcomes of four patients treated with five lesions were analyzed. Two of them had typical carcinoid tumors, and the other two had atypical carcinoids. All patients were not operated on due to comorbidities. All patients had 4D (four-dimensional simulation computer tomography scanning) performed while breathing freely. For tumors located more than 2 cm from the proximal bronchial tree (PBT), the dose of 54 Gy was administered in three fractions. For lesions located up to 2 cm from PBT, the dose of 50 Gy in five fractions was used. The median follow-up was 14.6 months. During this period, two of the patients were free from relapse. Massive dissemination to the liver was detected in one of the patients 2 months after the completion of SBRT. The changes in the liver were most likely present primarily—there is no precise information on the type of diagnosis used before radiotherapy. After about 12 months, the fourth patient underwent control PET examination, which ruled out recurrence of the disease [84]. The aforementioned analysis by Okoye et al. describes the results of a patient who underwent SFRT. He had a locally advanced disease of typical histology. The dose of 50 Gy in five fractions was used. In an evaluation of at least 3 months from the end of treatment, partial regression of the disease was described. Twenty-two months after the end of therapy, the disease showed permanent stabilization [66]. The team of Singh and colleagues conducted a retrospective analysis of the results of stereotactic/hypofractionated radiotherapy in 10 patients with 12 lesions of the carcinoid type [85]. Among them, nine patients had typical carcinoid and one atypical carcinoid. Patients with lesions smaller than 6 cm, general condition ≥70 according to the Karnofsky classification and without metastases beyond the thorax were qualified for treatment [85]. Most of the patients had CT performed on hold exhalation. Those unable to follow the protocol had 4D CT done. The GTV (gross tumor volume) area was defined based on the fusion of CT images with PET/CT. In patients who were treated on hold exhalation, PTV (planning target volume) was created by adding 11 mm up and down and 7 mm in other directions. For those who had 4D CT done to iGTV (total GTV resulting from GTV fusion contoured on CT from different respiratory phases) a 5mm margin was added to create a PTV. The median dose was 50 Gy (40–60 Gy), initially administered in 10 fractions, and as experience was gained, in five. Median BED was 100 Gy (75–115.5 Gy). The prerequisite for the approval of the treatment plan was at least 1000 mL volume of healthy pulmonary parenchyma and <12% of the lung volume receiving a dose >20 Gy. The median follow-up was 25.2 months (5.5–56 months). At follow-up, 2 out of 10 patients experienced lung or mediastinal lymph node progression. Median OS was 27.1 months (5.5–56 months). For patients with BED > 95 as well as those treated with five fractions, the median was 33.7 months, while with BED ≤ 95 and with 10 fractions, it was 20.5 months, respectively. It should be remembered that the patients in the study had comorbidities due to which they were disqualified from surgery [85]. Wegner et al. retrospectively analyzed the treatment results of 154 patients derived from the NCDB (National Cancer Database). Patients had T1-T2N0M0 stage and typical carcinoid tumor. Eighty-four patients (55%) received SBRT. The others received CFRT (conventionally fractionated radiation therapy). The median dose in the CFRT group was 54 Gy (50–60 Gy) in 24 fractions, while the median dose in the SBRT group was 50 Gy (50–55 Gy) in four or five fractions. The median follow-up was 30 months. There was an improvement in OS in patients receiving SBRT vs. CFRT (median 66 vs. 58 months). The summary presents the conclusion that SBRT can be considered the preferred form of therapy in inoperable patients [86].

## 5. The Role of Radiotherapy in the Treatment of LCNEC

The role of radiotherapy in neuroendocrine large cell carcinoma is not well defined and requires deeper analysis as well as further research. Rieber et al. performed a retrospective analysis of the treatment outcomes of 70 patients with LCNEC (most in stages IIA–IIIB). Overall, 93% of patients underwent surgery. Several stage IV patients with the oligomethatic phase of dissemination were also qualified for this procedure. Postoperative radiotherapy was administered to patients with N2 feature and incomplete resection, R1 and R2. The dose of postoperative radiotherapy was 50–60 Gy. In patients in higher stages, who could not undergo surgery, radiochemotherapy was administered at a dose of 66–70.5 Gy. The obtained results allowed for the conclusion that patients who underwent R1 or R2 surgery and underwent postoperative radiotherapy presented not worse results in terms of 2- and 5-year survival (50 and 30%, respectively, *p* = 0.89). It was also found that the OS of patients who underwent only surgery with R0 resection and those with adjuvant radiotherapy, radiochemotherapy or chemotherapy (mainly due to advancement ≥IIIA) did not differ significantly (*p* = 0.298). Similarly, there was no significant difference in local progression-free survival (lPFS) (*p* = 0.412). The researchers agreed that postoperative radiotherapy should be used in patients after R1 and R2 surgery as well as in pN2 patients. It was concluded that LCNEC is a radiosensitive neoplasm due to the comparable survival time of patients with R0 resection and those undergoing radiotherapy after R1 or R2 resection [87]. The team of Jiang et al. analyzed the treatment outcomes of 1619 patients with stage I-III LCNEC derived from the SEER database. We included 869 (53.7%) patients with stage I, 203 (12.5%) stage II and 547 (33.8%) stage III patients. The analysis of overall survival time (OS) and lung cancer-specific survival (LCSS) included patients who were operated on, who received only radiotherapy and were treated with surgery with adjuvant radiotherapy (PORT-postoperative radiation therapy). Stage I and II LCNEC radiotherapy did not improve OS and LCSS outcomes. However, in grade III, both OS and LCSS were significantly extended (*p* < 0.001). Interestingly, patients who were operated on and who underwent postoperative radiotherapy showed shorter OS (27 vs. 44 months, *p* = 0.012) and LCSS (37 vs. 93 months, *p* < 0.001). The median time of OS and LCSS after radiotherapy in patients who did not undergo surgery was longer, respectively (25 vs. 11 months, *p* < 0.001 and 34 vs. 12 months, *p* < 0.001). In patients who only underwent surgery compared to those who received postoperative radiotherapy, the OS and LCSS were longer, respectively (44 vs. 30 months, *p* = 0.024 and 93 vs. 38 months, *p* < 0.001). Summarizing the results, it was found that radiotherapy should be considered in LCNEC stage III, especially in patients who cannot undergo surgery. Combination therapy in the form of surgery with adjuvant radiotherapy was not recommended due to the shortening of the OS time [88]. Prelaj et al. analyzed the treatment results of 28 patients with stage III and IV LCNEC. The patients underwent six cycles of cisplatin-based chemotherapy with etoposide. Then, 10 of them received sequential radiotherapy on the thoracic lesions up to a dose of 60 Gy in 10 fractions. Patients who received thoracic radiotherapy (TRT) had a longer mOS survival of 28.3 months compared to 5 months in patients who did not receive TRT (*p* = 0.004). mPFS was also longer at 12.5 vs. 5 months (p = 0.02) [89]. According to the recommendations of the NCCN, this cancer is treated according to general recommendations for non-small cell lung cancer [24].

## 6. Conclusions

Despite the low percentage of neuroendocrine tumors in the population, especially lung lesions, it is extremely important to expand the clinical and practical knowledge with new methods of diagnosing and treating this type of cancer. Thanks to the constantly collected and updated data on the occurrence and treatment of NETs, NCCN (National Comprehensive Cancer Network) develops the latest guidelines for the management of this type of neoplasms for medical teams. The current recommendations presented by NCCN from 2020 indicate the use of stereotactic body radiation therapy (SBRT) for stage I and II advancement in all cases if the surgery is contraindicated or the patient does not consent to it. In tumors with low grading, high differentiation, the so-called in typical carcinoid tumors, in grades IIIA/IIIB/IIIC, radiotherapy or radiochemotherapy remains a therapeutic option and, according to some panelists, may be considered (evidence category 3). In the group of patients with intermediate grading, the intermediate type of differentiation, the so-called atypical carcinoids, radiotherapy or chemotherapy with cisplatin and etoposide or carboplatin and etoposide is generally recommended [24]. ENETS (European Neuroendocrine Tumor Society) is also involved in the process of developing guidelines and recommendations for the diagnosis and treatment of NET. According to their recommendations, the use of adjuvant treatment in patients with atypical carcinoids with lymph node involvement, especially in the presence of a high mitotic index, is highly recommended. However, they do not recommend adjuvant therapy for patients with typical carcinoid tumors [48,90]. In the last year, new guidelines related to the diagnosis, therapy and management of patients diagnosed with lung neuroendocrine neoplasm appeared, which also emphasize the importance of radiotherapy [21]. During the process of developing an effective strategy for the treatment of lung carcinoids, it seems important to study the recommendations of all centers in order to take the most favorable path of therapeutic management, contributing to an increase in the prognosis of treated patients. Due to the extremely significant diversity of all lung NET subtypes (based on clinical, histopathological or epidemiological differences), the selection of an appropriate therapeutic approach is an extremely difficult challenge for a multidisciplinary team of treating physicians including: gastroenterologist, pneumologist, endocrinologist, thoracic surgeons, pathologists, radiologists or nuclear medicine doctors, as well as oncological surgeons, clinical oncologists and radiation oncologists. That is why it is so important to conduct new prospective studies and therapeutic approaches to increase patients’ prognosis.

## Figures and Tables

**Figure 1 cancers-14-00177-f001:**
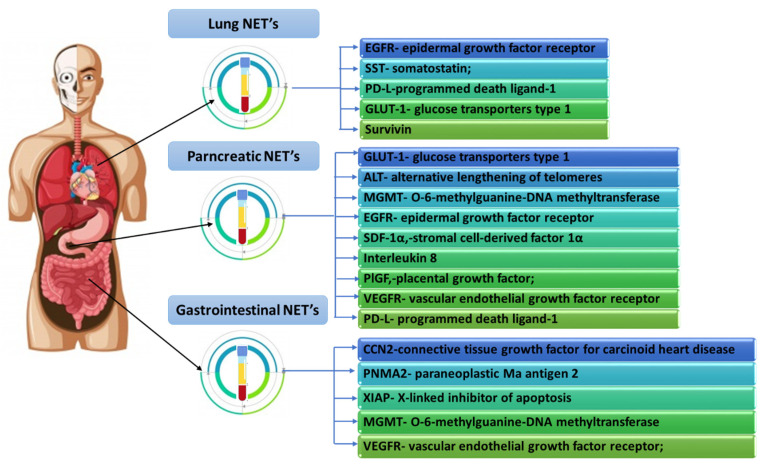
Selected examples of biomarkers determined in particular types of NET based on [39].

**Figure 2 cancers-14-00177-f002:**
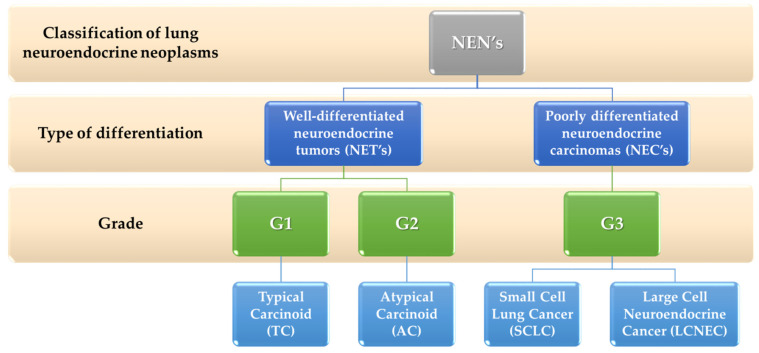
Division of lung neuroendocrine tumors based on [42].

**Figure 3 cancers-14-00177-f003:**
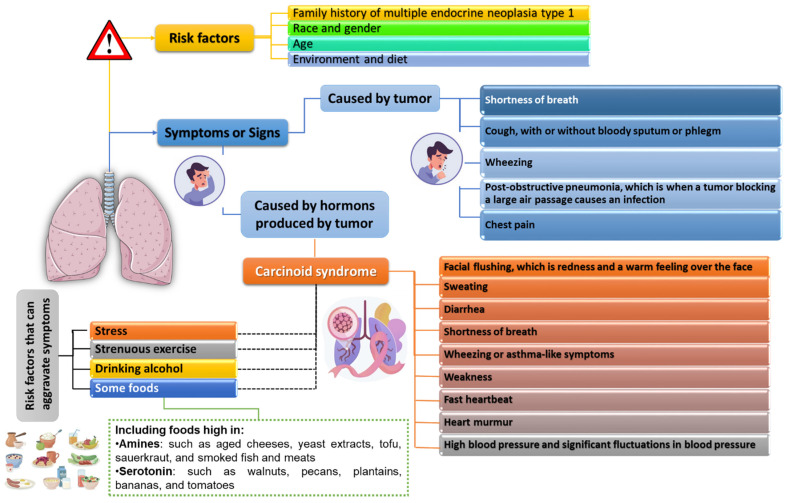
Risk factors, symptoms and signs of lung neuroendocrine tumors development based on [52,53].

**Table 1 cancers-14-00177-t001:** Characteristics of NET subtypes based on WHO classification and prognostic factors based on [29,30].

Characteristic	G1	G2	G3
Grade	Low	medium	high
Differentiation	Well	well	poorly
Ki-67 index [%]	≤2	3–20	>20
Mitotic index	<2/10 HPF	2–20/10 HPF	>20/10 HPF
Angioinvasion	Never	delay	always
Metastasis	−	−	+
Muscularis propia invasion	−	±	+
Prognosis	Slowly growing	Slowly growing	Agressive

**Table 2 cancers-14-00177-t002:** Characteristics of lung neuroendocrine neoplasms based on [50,51].

Characteristic	TC	AC	SCLC	LCNEC
Grade	Low	Intermediate	High	High
Morphological diversity	Well differentiated	Well differentiated	Poorly differentiated	Poorly differentiated
Common localization in the lung	central	peripherial	peripherial	Hilar/peripherial
Number of mitosis/2 mm²	<2	2–10	>10	>10
Presence of necrosis	Absence	Possible spot outbreaks	Very often on large fragments	Often on large fragments
Lymph node metastases at diagnosis	10–15%	50%	60–80%	60–80%
Distant metastases at diagnosis	3–5%	20–25%	40%	60–70%
Paraneoplastic syndrome	+	++	+	++++

Abbreviations: TC—typical carcinoid, AC—atypical carcinoid, SCLC—small cell lung cancer, LCNEC—large cell neuroendocrine lung cancer.

**Table 3 cancers-14-00177-t003:** Clinical advancement of lung neuroendocrine neoplasms based on the TNM scale according to the AJCC (American Joint Committee on Cancer) based on [62].

Grade	T Describes the Original Tumor	*N* Describes Whether or Not the Cancer Has Reached nearby Lymph Nodes	M Describes Whether There Are Distant Metastases
Latent cancer	Tx	*N*0	M0
Grade 0	Tis	*N*0	M0
Grade IA1	T1mi, T1a	*N*0	M0
Grade IA2	T1b	*N*0	M0
GradeIA3	T1c	*N*0	M0
Grade IB	T2a	*N*0	M0
Grade IIA	T2b	*N*0	M0
Grade IIB	T1a, T1b, T1c T2a, T2b T3	*N*1 *N*1 *N*0	M0 M0 M0
Grade IIIA	T1a, T1b, T1c T2a, T2b T3 T4	*N*2 *N*2 *N*1 *N*0, *N*1	M0 M0 M0 M0
Grade IIIB	T1a, T1b, T1c T2a, T2b T3 T4	*N*3 *N*3 *N*2 *N*2	M0 M0 M0 M0
Grade IIIC	T3 T4	*N*3 *N*3	M0 M0
Grade IVA	Any T Any T	Any *N* Any *N*	M1a M1b
Grade IVB	Any T	Any *N*	M1c

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
