# Peer review of "The Role of Conventionally Fractionated Radiotherapy and Stereotactic Radiotherapy in the Treatment of Carcinoid Tumors and Large-Cell Neuroendocrine Cancer of the Lung"

_cancers, 2021, doi:10.3390/cancers14010177_

Round 1

Reviewer 1 Report

Very interesting and impressive wotks summerazing a not well know disease.

Thank you for the review

The review is very exhaustive and clear, the figures are informative and beautiful. Tables are clear.

The main question is the state of the art of RT in case of treatment of carcinoid tumors and large cell neuro endocrine cancer of the lung.
It deals with the main point of the diagnosis of NET then it discuss the place of RT in case of TC and AT. This topic is original as it is a rare topic in clinical practice and a recent review can help physician to determine the care of the patient. Radiation oncologist will find here recommendations and reports from experiences to guide their practice and treatments.

Comparing to others published material is the fact that data are up to date and most recent articles are easier to find for clinicians. Moreover it highlights a rare topic in clinical practice. Methodology is not detailed according to PRISMA, however the literature is relatively poor and a detailed flow chart would not be really contributive to enhance the exhaustivity of the review. 

Conclusion summarize the role of RT in case of CT and AT carcinoids. More detailed data are found in the main text. The figures are beautify and informative, representations are very clear and I think it represents the results of a hard work. Tables are clear, and informative (Table 2 starts at the end of a page and could be moved to the top of the following one)
3 tables and 3 figures is enough to explain and to illustrate the main text of this original topic. 

Author Response

Dear Reviewer,

Thank you very much for the positive reviews of our work. As suggested, table 2 has been moved to the new page, so that its legibility and transparency is preserved.

Kind regards,

Paulina Niedźwiedzka-Rystwej

Reviewer 2 Report

This is an interesting overview regarding radiotherapy in neuroendocrine neoplasms of the lumg. Some important suggestions are based of the fact that the readers can be confused by reading epidemiology and characteristic features for both gastroenteropancreatic neoplasms and lung neoplasms. Please underline the fact of the rarity and focus the paper only in the lungs’ neoplasms. Please also describe the research strategy since it is not clear how you have selected the research papers which were included in this manuscript, which ones were exluded and why, whereras you have to add a prisma figure since it is very important to include all the papers that have been published till now in this issue.

Another important issue is that starting by the summary, we agree that carcinoid tumors and 3 large-cell neuroendocrine cancer of the lung need a multidisciplinary team, but the authors omitted a number of important disciplines as well. Gastroenterologists, Pneumologists, Endocrinologists, Thoracic Surgeons, Pathologists, Radiologists, Nuclear medicines doctors are of the specialties omitted since the 3 specialties they have reported cannot work independently but only as a team with the others. Please correct throughout the text.

The authors have to include the recently issued ESMO guidelines as well.

Lines 58-61; please do not repeat the rarity of the neoplasms but just add the references.

Lines 77-78; the reference 10 is too old to be placed here.

Lines 112-119 and Table 1 need review by a NET specialist

Author Response

Dear Reviewer,

Thank you very much for your time and effort put in the reviewing our paper. Please find our answers to your concerns below:

This is an interesting overview regarding radiotherapy in neuroendocrine neoplasms of the lumg. Some important suggestions are based of the fact that the readers can be confused by reading epidemiology and characteristic features for both gastroenteropancreatic neoplasms and lung neoplasms. Please underline the fact of the rarity and focus the paper only in the lungs’ neoplasms.

RE: Thank you very much for your comments on our work. When it comes to the issue of epidemiology and characteristics of both gastrointestinal and lung nauroendocrine neoplasms, a comprehensive approach to this issue seemed to be essential. Due to the prevalence of neuroendocrine neoplasms also in the digestive system, we have left this part in our review as a starting point for further analyzes. Therefore, we believe that the epidemiological characteristics of neuroendocrine neoplasms, due to their common occurrence also in the digestive system, in which radiotherapists are more often involved in treatment, should be included in a comprehensive manner.

Please also describe the research strategy since it is not clear how you have selected the research papers which were included in this manuscript, which ones were exluded and why, whereras you have to add a prisma figure since it is very important to include all the papers that have been published till now in this issue.

We started the collection of literature material for the creation of this review by searching for corresponding scientific articles in databases. In the Pub Med database, using the phrases to search for Radiotherapy lung carcinoid, 181 items were searched, from which the ones containing information on the technique of radiotherapy used, dosing schedule and fractionation of radiotherapy used, tumor advancement before radiotherapy and follow up results were selected. In the case of LCNC, 114 items were searched for using radiotherapy lung large cell neuroendocrine carcinoma. In addition, the literature of selected articles was also searched in the Google Scholar system. Due to the limited literature on the issue described by our team, we did not take into account the methodological strategy consistent with PRISMA. We thought that a detailed flow chart would not be really contributive to enhance the exhaustivity of the review.

Another important issue is that starting by the summary, we agree that carcinoid tumors and 3 large-cell neuroendocrine cancer of the lung need a multidisciplinary team, but the authors omitted a number of important disciplines as well. Gastroenterologists, Pneumologists, Endocrinologists, Thoracic Surgeons, Pathologists, Radiologists, Nuclear medicines doctors are of the specialties omitted since the 3 specialties they have reported cannot work independently but only as a team with the others. Please correct throughout the text.

Thank you very much for drawing our attention to other specialties that are also involved in the treatment of neuroendocrine neoplasms. We have placed an appropriate annotation of all of these specialties in the section devoted to introduction and conclusion.

The authors have to include the recently issued ESMO guidelines as well.

The appropriate annotation was placed in the sections introduction and conclusion.

Lines 58-61; please do not repeat the rarity of the neoplasms but just add the references.

As suggested, this has been corrected in the text.

Lines 77-78; the reference 10 is too old to be placed here.

As suggested, this has been corrected in the text.

Lines 112-119 and Table 1 need review by a NET specialist

As suggested, the text from the above-mentioned lines and Table 1 have been thoroughly checked by a NET specialist. The category of division into four groups has been removed from the text, and the section on treatment options has also been removed from the table.

 Once again, we would like to thank you for attention and kindness in reading our paper.

Kind regards,

Paulina Niedźwiedzka-Rystwej

Reviewer 3 Report

It is a very interesting review on a quite unexplored topic such as the use of radiotherapy for lung carcinoid and large-cell neuroendocrine tumors. The paper is well written and data are properly collected and presented. I suggest to include among the reference the following lung SBRT studies: 

  • Hypo-fractionated stereotactic radiation therapy for lung malignancies by means of helical tomotherapy: report of feasibility by a single-center experience. Radiol Med. 2018 Jun;123(6):406-414. doi: 10.1007/s11547-018-0858-7
  • Prognostic value of two geriatric screening tools in a cohort of older patients with early stage Non-Small Cell Lung Cancer treated with hypofractionated stereotactic radiotherapy. J Geriatr Oncol. 2020 Apr;11(3):475-481. doi: 10.1016/j.jgo.2019.05.002.

Author Response

Dear Reviewer,

Thank you for your favorable comments on our work. As suggested, both citations have been added to the publication- items 79 and 80.

Kind regards,

Paulina Niedźwiedzka-Rystwej

Reviewer 4 Report

I found your paper to be very interesting and informative. I think you have provided an easy to understand treatment guideline for a disease that is difficult to determine a treatment plan for in actual clinical practice.

I think the paper is fine as it is, but I would like to mention two parts that I think should be slightly revised.

1)

4.2. The abscopal effedt in the treatment of carcinoids

The Abscopal effect is very interesting and hopeful. However, it may be worth mentioning in the text that it is very rare.

2)

4.3. Stereotactic fractionated radiation therapy

397-398"Regardless of the type of cancer, low radiosensitivity is currently not a major problem for radiotherapy."

I thought this expression and a little exaggerated.
Even after stereotactic radiotherapy, residual lesions in the irradiated area can occur.

I do not think that this necessarily needs to be corrected, but I would appreciate it if it could be taken into consideration.

Author Response

Dear Reviewer,

Thank you for your time and effort devoted in correcting our paper. Here are the answers to your concerns:

I found your paper to be very interesting and informative. I think you have provided an easy to understand treatment guideline for a disease that is difficult to determine a treatment plan for in actual clinical practice.

I think the paper is fine as it is, but I would like to mention two parts that I think should be slightly revised.

  • 2. The abscopal effedt in the treatment of carcinoids

 The Abscopal effect is very interesting and hopeful. However, it may be worth mentioning in the text that it is very rare.

As suggested, we added a sentence in section 4.2 about the relevance of studies on the abscopic effect in terms of its rarity among patients with neuroendocrine lung cancer.

  • 3. Stereotactic fractionated radiation therapy

397-398"Regardless of the type of cancer, low radiosensitivity is currently not a major problem for radiotherapy."

This sentence has been removed from the work.

 I thought this expression and a little exaggerated.

Even after stereotactic radiotherapy, residual lesions in the irradiated area can occur.

 I do not think that this necessarily needs to be corrected, but I would appreciate it if it could be taken into consideration.

Thank you for drawing our attention to the important issues that has been corrected in the text. We do hope that the paper can be accepted in the corrected form.

Kind regards,

Paulina Niedźwiedzka-Rystwej